# A Simulation of the Real-Time Shelf Life of Frozen Fish Products in a Bulk System Sale

**DOI:** 10.3390/foods14081334

**Published:** 2025-04-12

**Authors:** Ilenia Dottori, Stefania Urbani, Luigi Daidone, Arianna Bonucci, Matteo Beccerica, Roberto Selvaggini, Beatrice Sordini, Raffaella Branciari, Gianluca Veneziani, Davide Nucciarelli, Agnese Taticchi, Maurizio Servili, Sonia Esposto

**Affiliations:** 1Department of Agriculture, Food and Environmental Sciences, University of Perugia, 06126 Perugia, Italy; ilenia.dottori@unipg.it (I.D.); stefania.urbani@unipg.it (S.U.); luigi.daidone@unipg.it (L.D.); m.beccerica@alice.it (M.B.); roberto.selvaggini@unipg.it (R.S.); beatrice.sordini@unipg.it (B.S.); gianluca.veneziani@unipg.it (G.V.); davide.nucciarelli@gmail.com (D.N.); agnese.taticchi@unipg.it (A.T.); maurizio.servili@unipg.it (M.S.); sonia.esposto@unipg.it (S.E.); 2Department of Veterinary Medicine, University of Perugia, 06126 Perugia, Italy; raffaella.branciari@unipg.it

**Keywords:** frozen fish, glazing process, quality evaluation of frozen fish, real-time shelf life, *Thunnus albacares*, *Merluccius hubbsi*

## Abstract

In this study, the real-time shelf life of frozen fillets of two different types of fish, tuna (*Thunnus albacares*) and hake (*Merluccius hubbsi*), was studied, simulating a bulk system sale. A glaze treatment was used on all the samples at the beginning, and during the 60-day storage period, the glaze was reapplied at regular intervals on half of the samples (“glazed”), while the other half was not re-glazed (“control”). To assess the quality changes in the two products, the peroxide value (PV), total volatile basic nitrogen (TVB-N), biogenic amines and volatile composition were determined every twenty days. Our results showed that both the glazed and control products did not exceed the legal limits of 35 mg/100 g of TVB-N and 100 mg/kg of histamine. In the hake fillets, in particular, glazing reduced the alteration phenomena associated with oxidative processes. In contrast, because of the high initial thickness of the glazing layer (20% of the weight of the frozen product), the reapplication of glazing during the storage period did not lead to any significant differences between the glazed and control tuna fillets. In conclusion, the different fishes’ compositions affect their conservation after the freezing process, which was improved by means of glazing in the case of the hake fillets.

## 1. Introduction

Fishes have high variability in their nutritional values, especially regarding their fatty acid profile [1]; in particular, they contain high amounts of proteins, lipids, fat-soluble vitamins and micronutrients. Fishes are also a very perishable food and are highly susceptible to oxidation and microbiological deterioration [2]. For this reason, effective preservation strategies are needed to increase their shelf life and ensure their safety and quality [3]. Once a fish has been caught, deterioration starts very quickly, with rigor mortis being the main culprit. At this stage, the metabolic activity of microorganisms and endogenous enzymes (autolysis) and the chemical oxidation of lipids lead to the degradation of chemical components and the formation of new ones, which are responsible for changes in safety and sensory quality [2]. Lipid oxidation is one of the main processes causing the quality deterioration of fish products, promoting free radical accumulation and rancidity [4]. It consists of a complex chain of reactions that can be divided into three phases: the primary phase, i.e., the formation of hydroperoxides; the secondary phase, in which the formation of hexanal and malondialdehyde occurs; and the tertiary phase, in which several molecules undergo decomposition or react with other molecules, and new negative compounds are formed [2,5]. Microbial development is the main mechanism of the oxidative deterioration of fish [6], with lipids being oxidised by both enzymatic and non-enzymatic means [7]. Fish fats are more susceptible to oxidation, both because of their higher content of polyunsaturated fatty acids [8] and due to rapid microbial growth, which promotes the production of volatile amines, biogenic amines, organic acids, sulphides, alcohols, aldehydes and ketones [9]. Refrigeration, deep-freezing and freezing techniques allow for fish to be stored for relatively longer periods without significant changes in quality [2]. However, lipid oxidation does not stop during frozen storage [10,11]. Refrigeration (storage at 0–4 °C) cannot guarantee long preservation times for fish [12]. Super-freezing, adopting temperatures between refrigeration and freezing, allows for the freezing of only 5–30% of the water contained [3]. During freezing, temperatures between −18 °C and −40 °C are used, and the process can be characterised as slow or fast deep-freezing [13]. With freezing, all microorganisms cease developing but can still remain in a dormant state. Indeed, freezing has a bacteriostatic effect [14]. To allow for further flesh preservation, glazing is carried out, covering the frozen products with a thin layer of ice (usually ranging between 5 and 15%, up to 50%, of the fish weight). In fact, during frozen or cold storage, seafood products may undergo surface drying and dehydration, which may lead to freezer burn and quality losses due to oxidation or rancidity. The potential and unique advantage of ice coating is its ability to exclude air from the surface of the product, preventing oxidation and thus extending the shelf life of the food [15]. Another advantage of this technology is that it is not expensive. In conclusion, the combination of freezing and glazing with ice is currently among the most extensively used methods for preserving fish and fish products in their natural state [16].

To the best of our knowledge, this is the first article focusing on the impact of different factors, such as light and oxygen, on frozen and glazed fish products over time, evaluating important safety and quality parameters during a real-time shelf life study. In order to fully assess the effect of these conditions, an ordinary sales condition of unpackaged fishes was simulated using two very different types of fish in terms of lipid contents: yellowfin tuna (*Thunnus albacares*, cut into slices) and Atlantic hake or Atlantic cod (*Merluccius hubbsi*, cut into fillets). During the 60 days of storage, sampling the products every 20 days, we analysed the peroxide values, total volatile basic nitrogen (TVB-N) content, biogenic amines and volatile compounds, which are considered indicators of deterioration in the quality and safety of fish products [2]. In particular, this study aimed to assess whether the process of re-glazing fish fillets could be strategically used to improve the shelf-life of the product, and therefore whether it was a technique that could be exploited by large-scale retailers.

## 2. Materials and Methods

### 2.1. Tuna

*T. albacares* (yellowfin tuna, belonging to the order *Perciformes* and family *Scombridae* [17]), was fished in the eastern–central Atlantic Ocean—FAO 34—with purse seines and harvesting nets and treated using IQF freezing after skin and bone elimination and slice cutting. The frozen slices were placed in plastic bags into boxes with a 6 kg capacity. Each box weighed 4.8 kg. The freezing date was 1 August 2020, whereas the packaging date was 16 March 2020. The relative “Best if Used by Date” was 1 August 2023. The label nutrition table per 100 g of product was as follows: energy value: 439 kJ/105 kcal; fat: 0.2 g; relative saturated fatty acids: 0.06 g; carbohydrates < 0.1 g; relative sugars: <0.1 g; proteins: 26 g; salt: 0.88 g.

### 2.2. Hake

*M. hubbsi* (Atlantic hake or Atlantic cod, belonging to the order *Gadiformes* and family *Merlucciidae* [17]), was fished in the southwest Atlantic Ocean—FAO 41—using trawl nets and frozen on board using the interleaved technique after skin and bone elimination and fillet cutting. Each fillet weighed 80–120 g. Thus, the gross weight of the carton was 14 kg, inside which were two interleaved blocks of fillet, each weighing 7 kg, giving a net weight of 13.3 kg. The fish were caught on 23 October 2021, which was also the freezing date; the date by which they must be consumed was 22 October 2023.

### 2.3. Glazing Treatment

The product was placed in perforated trays, which were quickly immersed in a bath of drinking water and ice at a temperature of 0–2 °C. The temperature was constantly measured with a probe thermometer and more ice was added if the temperature exceeded 2 °C. After a few seconds of immersion, the product was drained and placed in a cold store at −25 °C. The operation was very quick and the layer of ice added was 1–2%. Finally, the product was put back on sale.

### 2.4. Shelf Life Trial

At time T0, fillets of yellowfin tuna and Atlantic hake were glazed, as reported in Section 2.3, placed in frozen boxes and stored at −25 °C. Subsequently, both the tuna and hake fillets were placed inside open freezers (−18 °C) and exposed to LED illumination for 11 h per day for 60 days, simulating a commercial shelf life. Periodically, half of each product was subjected to glazing reapplication (“glazed” samples) while the other half was not (“control” samples). Over this period, the fillets were carefully moved into the freezer to guarantee equal exposure to light.

### 2.5. Sampling and Preparation for Analysis

Over the 60 days of the real-time market shelf life test, samples (300 g each) from the tuna and hake fillets with and without reapplied glazing were withdrawn every 20 days. Fish samples were collected at T0, T1 (after 20 days), T2 (after 40 days) and T3 (after 60 days), homogenised using an electric grinder MQ30 (Braun, Kronberg im Taunus, Germany) and stored at −80 °C prior to analysis.

### 2.6. Determination of Peroxide Value

An amount of 50 g of tuna and 120 g of hake fillets were homogenised with 200 mL of a chloroform/ethanol (Carlo Erba Reagents, Milan, Italy) mixture (1:1 *v*/*v*). The resulting mixture was sonicated in an ultrasonic bath (VWR, Milan, Italy) at 25 °C for 30 min. Subsequently, the sample was stirred at 215 rpm for 30 min at 25 °C and then filtered through Perfect 2 filter paper (Cordenons, Milan, Italy). The filtrate was evaporated under vacuum at 35 °C until the solvent was completely evaporated. The peroxide value of the residual oil was determined in accordance with EU Reg. No. 2019/1604 [18].

### 2.7. Determination of Total Volatile Basic Nitrogen (TVB-N)

The contents of TVB-N were determined according to the reference method described in EC Reg. No. 2005/2074 [19] by applying an automatic distillation unit with a titrimeter (UDK149, VELP Scientifica, Usmate Velate, Italy).

### 2.8. Determination of Biogenic Amines

The extraction of biogenic amines was conducted as reported by Molognoni et al. [20], with the following modifications: the hake and tuna fillet samples (2 g) were homogenised using Ultra Turrax (VWR, Milan, Italy) for 1 min at 15,000 rpm, with 10 mL of acetonitrile/methanol/acidified water with a 0.1% formic acid (45:45:10 *v*/*v*) solution; the resulting mixture was sonicated in an ultrasonic bath (VWR, Milan, Italy) for 20 min at 35 °C and then centrifuged at 6000 rpm for 10 min. The supernatant was recovered, and the volume was reduced to 5 mL using rotavapor at 37 °C and stored at −80 °C for 12 h. For the quantitative analysis of biogenic amines, 1 mL of the frozen extract was filtered through a 0.22 µm PVDF syringe filter (Carlo Erba Reagents, Milan, Italy), and 10 µL of the filtered sample was injected into an ultra-high performance liquid chromatography/quadrupole time-of-flight mass spectrometry (UHPLC/Q-TOF-MS) instrument. A Zorbax Eclipse Plus C18 100 mm × 2.1 mm with a 1.8 μm column (Agilent Technologies, Santa Clara, CA, USA) was used for the separation of the biogenic amines while the elution was performed at a flow rate of 0.3 mL/min using water that had been acidified with 0.1% formic acid (Carlo Erba Reagents, Milan, Italy), which was solvent A. Acetonitrile (Carlo Erba Reagents, Milan, Italy) that had been acidified with 0.1% formic acid was solvent B. The elution gradient was varied as follows: at time 0 min, the solvent composition was 90% A and 10% B. This was maintained for 1 min, after which it was changed to 80% A and 20% B. This composition was maintained for 8 min. After 5 min, the composition was changed to 50% A and 50% B, and after a further 5 min, it was changed to 0% A and 100% B; this composition was maintained for 5 min. The initial conditions were restored and the system was allowed to equilibrate for a further 5 min. The total duration of the analysis was, therefore, 30 min, while the acquisition time was 25 min. The mass spectrum was measured by means of ESI ionisation in positive mode in the *m*/*z* range of 40–1700 with a scan rate of 1.5 spectra/s, simultaneously infusing; this is in addition to the eluent from the HPLC system (via the first nebuliser), the two reference masses (via the second nebuliser) having *m*/*z* 121.050873 and 922.009798. The dual ESI source parameters were as follows: gas temperature, 325 °C; drying gas flow, 11 L/min; nebuliser pressure, 35 psig; capillary voltage, Vcap 4000 V; Fragmentor, 110 V; skimmer, 65 V; octapole 1 RF, 750 V. Data were acquired in MS/MS mode by acquiring the chromatogram in MS/MS by selecting the ions using the quadrupole. The Agilent MassHunter B. 10.00 software (Agilent Technologies, Santa Clara, CA, USA) was used to perform the analysis and to identify and quantify the compounds. For the quantitative evaluation of biogenic amines, calibration straight lines were constructed for each compound in the concentration range of 0.2 mg/L–20 mg/L. The results were expressed in mg/kg.

### 2.9. Determination of Volatile Compounds

The composition of volatile compounds was determined by means of gas chromatography-coupled mass spectrometry (GC/MS) analysis based on headspace sampling and using Solid Phase Microextraction (SPME). The samples of fish fillets (1 g) were placed in 20 mL vials and spiked with 2 mL of a saturated NaCl solution and 50 μL of an internal standard (4-methyl-2-pentanol, 750 μg/L), sealed and placed in the autosampler. Subsequently, SPME sampling of volatile compounds was performed by exposing the fiber consisting of Carboxen/divinylbenzene/polydimethylsiloxane 50/30 µm, which was 2 cm long (Supelco Inc., Bellefonte, PA, USA). Before adsorption, the sample was kept under stirring (750 rpm) for 30 min at 40 °C. The adsorbed compounds were then thermally desorbed for 5 min by inserting the fiber into the GC injector and maintained at 250 °C. The analyses were carried out with an Agilent Technologies 7890B GC, equipped with a “Multimode Injector” (MMI) 7693A (Agilent Technologies, Santa Clara, CA, USA) and a thermostat PAL3 RSI 120 autosampler equipped with a fiber conditioning module and shaker (CTC Analytics AG, Zwingen, Switzerland); this was coupled to a single quadrupole MS 5977B (MSD) with an XTR (Extractor Ion Source) electron impact source (Agilent Technologies, Santa Clara, CA, USA). For the separation of the volatile compounds, a DB-WAXetr fused silica capillary column was used, with a length of 50 m, an i.d. of 0.32 mm and a film thickness of 1 µm (Agilent Technologies, Santa Clara, CA, USA). Helium was used as carrier gas at a flow of 1.7 mL/min, which was kept constant throughout the analysis time by means of an Electronic Flow Control (EFC) device. The GC column oven temperature was set according to the following schedule: the initial temperature was 35 °C and maintained for 4 min, then increased to 45 °C at 5 °C/min, further increased to 150 °C at 4 °C/min, further increased up to 180 °C at 8 °C/min, maintained for 2 min and, finally, brought to 210 °C at 11 °C/min and maintained for 13.77 min; under these conditions, the total analysis time was 55 min. The injector was set at a temperature of 250 °C. The temperature of the “transfer line” was set at 215 °C; regarding the experimental conditions of the mass spectrometer, the temperature of the source was 190 °C and that of the quadrupole was 150 °C. The electron impact mass spectrum (EI) was recorded with an ionisation energy of 70 eV in the mass range 25–350 a.m.u., with 4.3 scans/s, and the MS spectra were acquired in scan mode. The processing of the collected data was carried out using the Agilent MassHunter B.08.00 software (Agilent Technologies, Santa Clara, CA, USA) with the Unknown Analysis module. The identification of volatile compounds was performed by comparing the mass spectra and retention times thus obtained with those of pure analytical standards and with the spectra of the NIST-2014 library. The volatile compounds were quantified and expressed in µg/kg by comparing the area of each peak of the extracted ion (corresponding to each compound evaluated) with the ion area of the peak of the internal standard (4-methyl-2-pentanol), as reported by Dottori et al. [21].

### 2.10. Statistical Analysis

To compare the results obtained in this experiment and test the differences between the different storage times, a one-way ANOVA was performed using Tukey’s test, while the *t*-test was used to assess the differences between the different treatments at the same storage time. Both univariate statistical analyses were performed using SigmaPlot v.12.3 software (Systat Software, Inc., San Jose, CA, USA). Furthermore, the entire dataset obtained from the instrumental analyses of the tuna and hake fillet samples was analysed using the multivariate statistical technique of Principal Component Analysis (PCA). The data were first normalised and auto-scaled to give a zero mean and a unit standard deviation for each variable. Cross-validation was also used to determine the number of significant components in the model definition. All multivariate statistical processing was conducted using the chemometric package SIMCA v.13.0.3.0 (Umetrics AB, Umeå, Sweden).

## 3. Results and Discussion

### 3.1. Results of Statistical Analysis

#### 3.1.1. PCA of Hake

The Principal Component Analysis (PCA) applied to the results of our analysis of the hake samples explained 93% of the total variance with three significant principal components (explaining 80%, 8% and 5%, respectively). The relative score plot (Figure 1a) of the first two principal components shows a separation of the samples according to the time of storage along the first component (from the left to the right side of the score plot) and a differentiation of the fillets according to the treatment (glazed vs. control) along the second component (from the lower to the upper side of the score plot). In this regard, it can be seen from the loading plot (Figure 1b) that all samples were characterised by higher concentrations of acetoin at T0. As the days of storage progressed, an accumulation of volatile substances related to the oxidation of fats (such as butanal, pentanal, pentanol, 1-octen-3-ol and nonanal), as well as histamine and TVB-N, was observed in the non-glazed hake fillets. This result might indicate that the glazing process protected the fillets against both photo-oxidation and microbial activity. Indeed, the glazed ice layer excludes air from the surface of the product, thus reducing the rate of oxidation [15]. In score plot t1/t3 (Figure 1c), we observed a differentiation of objects according to their treatment along the third component for all storage times, and the glazed samples were positioned lower than the controls. The loading plot (Figure 1d) showed that the control samples had higher contents of acetoin, 1-octo-3-ol, pentanol and peroxide, while the glazed samples had higher contents of cadaverin, 2-methylbutane and 3-pentanol.

#### 3.1.2. PCA of Tuna

The PCA model for the tuna explains 95% of the total variance, with three significant principal components (each explaining 80%, 8% and 7%, respectively). The score plot of the first and second principal components (Figure 2a) showed a distribution of the samples according to conservation (from the left to the right side of the score plot), irrespective of the treatment that they underwent. Along the second component, the control samples were always positioned higher than the glazed ones. From the corresponding loading plot (Figure 2b), it can be seen that in both samples (control and glaze), as the duration of storage increased, the content of 3-methylbutanal, 2,3-pentanedione, hexanal, the two biogenic amines (histamine and spermidine) and (E,E)-3,5-octadiene-2-one increased.

In score plot t1/t3 (Figure 2c), a clear separation of the samples can be seen along the third main component in relation to the treatment (control vs. glazed). The glazed samples were distributed downwards and mostly had higher contents of octanal, 2,4-heptadienal, propanal and dimethyl sulphide. The control samples, in contrast, were distributed above these and showed higher values of 1-penten-3-ol, TVB-N, pentanal, nonanal and 2-methyl-butanal.

These preliminary results suggest that the glazing process, in addition to temperature and storage time, could positively influence the quality characteristics of both fish throughout their shelf life. Further analyses were therefore conducted in order to assess the development of several parameters.

### 3.2. Peroxide Value (PV)

The peroxide value is a parameter for determining the degree of lipid oxidation [22], including in fish products. During the storage of a frozen product, lipid oxidation proceeds more slowly, and therefore, fewer oxidation products are formed. In fish, this also depends on the initial lipid content, temperature and length of the storage period [23].

Peroxides are used to measure the primary products of lipid oxidation, in particular, hydroperoxides. They undergo decomposition by reducing the peroxide value and give rise to a wide variety of aldehyde control molecules [24]. Our analyses showed an increase in these molecules during exposure to the tested products.

Table 1 shows that in the hake, the peroxide value (PV) at T0 was 0 but it increased significantly during the 60-day trial to 12.6 ± 0.3 meq O_2_/kg. Meanwhile, in the glazed sample, no peroxides were detected until the 60th day of experimentation, where they reached a value of 2.6 ± 0.1 meq of O_2_/kg. The differences between the two treatments were statistically significant, which showed that glazing had a protective effect against lipid oxidation. These results were consistent with those reported in other studies; e.g., by Nowsad et al. [23], who studied frozen cubes of Hilsa shad (*Tenualosa ilisha*) stored for 6 months and reported an increase in PV from 1.80 to 9.30 meq O_2_/kg. An increase in peroxide value over time was also observed by Aberoumand and Baesi [25] in *Lethrinus atkinsoni* that was stored at −18 °C for 40 days; in this case, the PV increased from 1.29 ± 0.08 to 2.35 ± 0.03 meq of O_2_/kg. The importance of storage conditions was also demonstrated in the study by He et al. [22], where in Tilapia fillets, the initial PV of around 0.76 meq O_2_/kg increased to 3.69, 2.35 and 1.48 meq O_2_/kg after 40 days at −4 °C, −8 °C and −18 °C, respectively. Similar results were also obtained in several other studies [24,25,26,27,28,29].

In tuna, the peroxide value at T0 was 14.0 meq of O_2_/kg, which was very high compared with the hake. However, this could be attributed to several factors, such as the higher fat content in tuna flesh, as reported by Nowsad et al. [23], and the longer storage time before the study (the hake was caught on 23 October 2021, while the tuna was caught on 1 August 2020). It was therefore possible to assume that because the fish had very different characteristics, the PVs of their flesh were naturally different. At T1 (20 days), in the control and glazed tuna, there was a significant increase in PV of 54.3% and 56.4%, respectively, compared with the initial value, reaching values of 21.6 ± 1.1 and 21.9 ± 0.8 meq O_2_/kg. This increase aligns with the studies mentioned above. There were no significant differences between the two treatments (control vs. glazed), which can be explained by the large amount of icing (20%) that was used on the tuna at the beginning. From T1 to T3, there was a decrease in PV in both cases. This was also to be expected based on Chaijan’s report [24], which showed that the decrease in PV at the end of salting occurred at the same time as the increase in TBARS. They also reported that the decrease in PV was probably due to the decomposition of hydroperoxides. In fact, hydroperoxides decompose in several steps, resulting in a variety of secondary oxidation compounds, including aldehydes [24]. The decrease was 34.6% in the control sample and 45.6% in the glazed sample, with no significant differences between the two treatments. This was, again, probably due to the thick glaze. The values went from 21.6 ± 1.1 to 14.2 ± 0.4 meq O_2_/kg and from 21.9 ± 0.8 to 11.9 ± 0.5 meq O_2_/kg in the control and glazed samples, respectively. This reduction in PV was also in agreement with studies by Aberoumand and Baesi [25], Chaijan [24], Rezaei et al. [29], Santos et al. [26] and Iglesias et al. [27]. In the study by Rezaei et al. [29], which examined rainbow trout that were kept on ice for 20 days, there was an initial increase in PV from around 1.5 meq O_2_/kg up to a maximum of 10 meq O_2_/kg after 8 days and then a decrease to 2.5 meq O_2_/kg on day 20. In Chaijan’s [24] study on thawed Tilapia, focusing on the salting process, the PV decreased from around 21 to 5 meq of O_2_/kg in just 30 min as a result of the room temperature, accelerating the peroxide degradation processes. Our tuna also exhibited a more intense reduction in PV than in other studies, probably as a result of the extreme storage conditions to which it was exposed. However, an increase in the secondary compounds of oxidation was observed based on the analysis of volatile compounds. It is fair to assume that if the study had continued for longer, the PV of the hake would also have decreased, which is an aspect that requires further investigation.

### 3.3. Total Volatile Basic Nitrogen (TVB-N)

Total volatile basic nitrogen (TVB-N) is a spoilage index for determining the freshness of fish; the legal limit is 35 mg/100 g [30]. The production of TVB-N is related to the metabolism of spoilage bacteria and the activity of endogenous enzymes that cause the degradation of proteins and non-protein nitrogen compounds, generating volatile ammonia control compounds (NH3), trimethylamine (TMA) and dimethylamine (DMA). These compounds are responsible for the unpleasant odour. The increase in TVB-N is also caused by the autolytic deamination of amino acids [31]. Li et al. [32] conducted a study on common carp (*Cyprinus carpio*) and reported that each type of fish had a different TVB-N value as a result of endogenous and exogenous factors and that even within the same fish species, it could be different, e.g., between white and dark muscles. During frozen storage, there was a slow and steady increase in both types of muscle, with white reaching a higher value. In general, TVB-N showed a constant increase during storage depending on the temperature [16,22,31,33,34,35,36]. Table 2 shows the TVB-N data.

Based on our analyses, the initial and final TVB-N values of both hake and tuna did not exceed the legal limit of 35 mg/100 g. The hake had an initial value of 16.8 mg/100 g. A similar value (16.41 mg/100 g) was found in fresh sea bass fillets in a study by Messina et al. [34]. The TVB-N value slowly increased over 60 days in both the control and glazed fish, reaching 24.5 ± 2.3 and 19.6 ± 1.7 mg/100 g, respectively. There were no significant differences between the two treatments. This could be attributed to the fact that in a frozen stored product, microbial and enzymatic activity are minimised. Luo et al. [16] conducted a study on Pacific saury (*Cololabis saira*), comparing a simple glaze with one containing natural control substances such as nisin, chitosan and phytic acid. Fishes stored at −18 °C for 12 months and with antimicrobial substances present in the glaze had a significantly lower TVB-N content. This confirms that it was necessary to treat the microorganisms in order to reduce the TVB-N content and that a simple layer of ice was not sufficient. This was also confirmed by other studies, such as those by He et al. [22], Messina et al. [34] and Ribeiro et al. [37]. Luo et al. [16] also specified that icing and freezing are widely used methods for fish preservation; however, during frozen preservation, a decrease in fish freshness may occur due to slow microbial growth and enzyme activities in muscle tissues. Glazing is therefore rather inefficient for long-term storage, especially for certain fatty fish species [16].

The same was found for tuna, which started from a TVB-N value of 20.3 mg/100 g and slowly increased over the 60 days in both the control and glazed samples to 25.2 ± 2 mg/100 g and 22.4 ± 2.1 mg/100 g, respectively, representing increases of 24.1% and 10.4%; however, the data were not significant, and there were no significant differences between the two treatments. The different starting values and increases between tuna and hake were probably attributable to intrinsic and extrinsic factors.

### 3.4. Biogenic Amines

Microbial activity is responsible for the production of biogenic amines, such as histamine, cadaverine, tyramine and putrescine [9]. The production of biogenic amines can be induced by the decarboxylation of the corresponding free amino acids by bacteria such as *Shewanella putrefaciens*, *Hafnia alvei* and *Morganella morganii* [12]. Some factors that influence microbial and enzymatic activity are temperature, pH, water activity and oxygen availability. The combination of these factors can be responsible for the variability of biogenic amine content. The prevention of biogenic amine formation in raw fish is mainly based on rapid chilling after catch and subsequent storage at ice-cold temperatures, as well as good handling and hygiene practices on board vessels. Ice, slurry ice or mechanically chilled seawater can be used to chill fish after harvesting [38]. One of the most important biogenic amines is histamine; EU Reg. No. 2005/2073 sets a legal limit for its content of 100 mg/kg [39]. Histamine is a very toxicologically stable molecule and causes scombroid syndrome [12]. Fish of the scombroid family are commonly involved in the onset of this intoxication due to the high levels of free histidine in their muscle tissue [31]. When a fish is caught, its histamine content is very low but tends to increase gradually. Several authors have reported a very high level of histamine in fish that had undergone prolonged storage at room temperature, whereas lower values were found in fish that were stored frozen, as the reactions were slower. Table 3 shows the data for biogenic amines based on our analyses.

In this study, histamine was found in both fishes, while cadaverine was found in hake and spermidine in tuna. The histamine values that were reached at the end of the trial (60 days) were below the legal limit. Dalgaard et al. [40] described the importance of storage conditions in the production of biogenic amines, in particular, temperature and time. Emborg et al. [41] and Dalgaard et al. [40] described that the production of different amines depended on the type of microbial flora that were present in the fish. Dalgaard et al. [40] and Kung et al. [31] reported that the production of biogenic amines was also related to the composition of the fish flesh. This explains why the amines found in the two fishes were not the same, except for histamine, which had very different values. In hake, cadaverine was only found after 40 days of exposure, with a value of 0.6 mg/kg, which rose to 0.8 mg/kg after 60 days, but the increase was not significant. In the glazed product, on the other hand, cadaverine was only found after 60 days, and there was no significant difference with the control. The histamine content in the control was 0.4 mg/kg and 1 mg/kg, respectively, at 40 and 60 days, with a significant increase. In the glazed sample, only 0.2 mg/kg was measured after 60 days, there was a significant difference with the control: this allowed us to conclude that the glaze played a fundamental role in inhibiting histamine formation. This is confirmed by the study by Gandotra et al., which shows that the microbial load of the ice-glazed Rohu fillets was significantly lower than that of the control samples [42]. This finding was supported by the study of Dalgaard et al. [40], in which an experiment was carried out on garfish (*Belone belone*) that were stored under refrigeration in a modified atmosphere. It was shown that in most cases, modifying the products’ atmosphere or limiting contact with the air reduced histamine production. Kung et al. [31] conducted a study on Milkfish sticks (*Chanos chanos*) that were stored at different temperatures—under vacuum and in plain PET—and found that the limiting air contact reduced histamine production, thus limiting microbial development.

Histamine and spermidine were already present in tuna at T0. Spermidine increased in the control sample from 1.1 mg/kg to 12.1 mg/kg after 60 days (+1025%). In the glazed tuna, from a starting value of 1.1 mg/kg, it increased to 11.8 mg/kg (+99%). There were no significant differences between the two treatments (control vs. glazed). This result, as already mentioned, could be attributed to the 20% initial glazing. This trend was not in accordance with the results found by Cattaneo [43], who reported a decrease in spermidine during storage. However, it has been demonstrated that not all fish species have followed this trend. The histamine content in the control started at 4.1 mg/kg at T0 and reached 15.7 mg/kg at 60 days, representing a significant increase of 279%. In the glazed sample, a value of 12.3 mg/kg was reached on day 60, representing a significant increase of 198%. There were no significant differences between the two treatments. However, higher values of biogenic amines were observed in the tuna than in the hake. For histamine, according to Kung et al. [31], this was due to the composition of tuna meat, which is richer in the amino acid histidine.

### 3.5. Evolution of Volatile Compounds

Volatile compounds can be generated by enzymatic reactions, lipid oxidation or microbial action. The action of microorganisms is irrelevant during storage at freezing temperatures [27]. However, freezing cannot prevent lipid oxidation, leading to the formation of volatile substances [44]. The oxidation of lipids is commonly expressed by the peroxide value (PV) and TBARS. However, as peroxides decompose into oxidation by-products and TBARS are not specific, the evaluation of volatile compounds is a popular means of assessing lipid oxidation in hake [27,28]. In general, the main substances formed are aldehydes, esters, ketones and alcohols [32]. According to Iglesias et al. [27], aldehydes are the compounds that increase the most during frozen storage, which was also found in this study. Of the aldehydes, hexanal is the one that is most commonly present: it is formed by means of the oxidation of linoleic acid, which is very abundant in fish muscle, or through the degradation of the preformed volatile compounds control 2-octanal. As for alcohols, we observed that their concentration also tended to increase during frozen storage, particularly 1-penten-3-ol and 1-otten-3-ol, which was in accordance with the study by Iglesias et al. [27]. In that study [27], an increase was also found in certain ketones, such as 2,3-octanedion and 2,3-pentanedion; this increase was also observed in our study but only for 2,3-pentanedion. Figure 3a–h shows the characteristic volatile compounds of hake and tuna.

In the hake, all the detected aldehydes exhibited an increasing trend. The sum of the aldehydes in the control at T0 was 56.5 µg/kg, and at T3, it was 150.3 µg/kg, representing an increase of 166.0%. In the glazed sample, there was a significant increase between T1 and T2 (+45.0%). There was a significant difference between the control and glazed samples at T3, which shows that glazing reduces lipid oxidation. There was an increase in propanal in the control from 45.6 µg/kg at T0 to 107.4 µg/kg at T3 (+135.5%). In the glazed sample, a value of 64.7 µg/kg was reached at T3, representing an increase of 41.9% from T0. Hexanal in the control increased from 3.8 µg/kg at T0 to 10.2 µg/kg at T3, representing an increase of 168.4%. In the glazed sample, at T3, a value of 6.1 µg/kg was reached, representing a significant increase of 60.5% from T0. For both molecules, when comparing the two treatments, there was a significant difference at T3. In the control, the content of (E,E)-2,4-heptadienal increased by +128.2% from T1 to T3, while in the glazed sample, it was only found at T3. In the tuna, all aldehydes exhibited an increasing trend, as they did in hake, but the values were higher, probably due to the different compositions. The sum of the aldehydes at T0 was 933.8 µg/kg, while it was 1372.4 µg/kg at T3, representing a significant increase of 47.0%. In the glazed sample, 1337.6 µg/kg was reached, representing an increase of 43.2% from T0. In terms of the sum of aldehydes, there were no significant differences between the two treatments. In tuna, the main aldehyde was hexanal, which started at 477.0 µg/kg in the control and reached 706.9 µg/kg, an increase of 48.2%. In the glazed sample, the value reached 695.6 µg/kg, representing an increase of 45.8%. For this compound, there were no significant differences between the two treatments. Propanal in the control rose from 156.7 µg/kg to 244.2 µg/kg, representing an increase of 55.8%, which was significant between T0 and T2. In the glazed sample, the value of 235.4 µg/kg was reached, representing a significant increase of 50.2% between T0 and T1. There were no significant differences between the two treatments. (E,E)-2,4-heptadienal did not increase significantly in either the control or the glazed sample, and there were no significant differences between the two treatments.

The alcohols in the hake showed a constant increase over time. The sum of alcohols in the control was equal to 146.1 µg/kg at T0 and reached 228.5 µg/kg at T3, representing an increase of 56.4%. In the glazed sample, it reached 162.6 µg/kg, representing an increase of 11.3% from T0 but this was not statistically significant. Comparing the two treatments, the difference was significant at T3. In the tuna, the sum of alcohols showed an increasing trend: in the control it was 3343.4 µg/kg at T0 and reached 3907.5 µg/kg at T3, representing an increase of 16.9%. In the glazed sample, it reached a value of 3692.2 µg/kg, representing a non-significant increase of 10.4% between the different times. There were no significant differences between the two treatments.

In terms of the sum of ketones, in the control, it increased from 10.9 µg/kg at T0 to 15.9 µg/kg at T3, representing a statistically significant increase of 45.9%. In contrast, there was an initial decrease in the glazed sample from T0 to T2 (−18.3%) but it was statistically non-significant; meanwhile, from T2 to T3, the sum of ketones reached a value of 10.5 µg/kg, representing a non-significant increase of 18.0%. Comparing the two treatments, the difference was significant at T2 and T3. In the tuna, the sum of ketones increased during the experiment, with the exception of acetoin, as in the hake. The sum of ketones at T0 was 370.0 µg/kg, and it increased to 500.2 µg/kg at T3, representing an increase of 35.2%. In the glazed sample, the value was 475.4 µg/kg at T3, representing a non-significant increase of 28.5%. There were no significant differences between the two treatments in tuna.

In the hake, the sum of the other compounds in the control at T2 had a value of 4.1 µg/kg, which increased to 4.2 µg/kg at T3. In the glazed sample, it was 3.3 µg/kg at T3. There were non-significant increases and differences between the two treatments. In the tuna, the sum of the other compounds in the control at T0 was 862.6 µg/kg, which reached a value of 1270.0 µg/kg at T3, representing an increase of 47.2%. This was only significant between T0 and T1. In the glazed sample, it reached 1231.3 µg/kg at T3, representing an increase of 42.7%, which was only significant between T0 and T1. We observed a difference between the control and glazed sample in hake, which was smaller than that observed in tuna, probably as a result of the 20.0% glazing. The increase in volatile compounds mirrors what was reported in the peroxide number analysis, as they are secondary and primary products of oxidation, respectively, which is also in agreement with Iglesias and Medina [27,28] and Leduc et al. [44].

## 4. Conclusions

Using multivariate statistical analysis, it was possible to distinguish between the samples of hake and tuna according to their storage time, as well as the different treatments that they underwent (with or without reapplication of the glaze). Both products, glazed and unrestored, did not exceed the legal limits for TVB-N (<35 mg/100 g) and histamine (<100 mg/kg) but remained far below these limits. The TVB-N content increased in both products but not significantly since microbial activity was inhibited at freezing temperatures. There were no significant differences between the glazed and control samples in both fishes. Biogenic amines, in particular histamine, increased significantly in both fishes during 60 days. In particular, in the hake, there was a significant difference between the glazed and control samples: after 60 days, the control had a histamine content of 1 mg/kg while the glazed sample had 0.2 mg/kg. In tuna, the histamine levels were higher than in hake but there were no statistically significant differences between the control and glazed samples. The peroxide value first increased significantly in both fish but after 40 days of storage, it decreased significantly in the tuna, indicating the production of secondary oxidation compounds. In the hake, after 60 days, there was a significant difference between the glazed (2.6 meq of O_2_/kg) and control samples (12.6 meq of O_2_/kg). In contrast, there was no significant difference in the tuna, probably as a consequence of the thick initial glazing layer (20%) on the control samples, thus subjecting them to very similar exposure conditions to the glazed samples. The volatile compounds in the hake, which were mainly represented by aldehydes, alcohols and ketones, exhibited significant increases towards the end of the trial, and there was a significant difference between the glazed and control samples. In the tuna, there was an increase in these substances over the course of the storage but there were no major differences between the glazed and control samples. In conclusion, the different compositions of the two fish significantly influence their alteration. In the hake, the reapplication of the glaze played a key role in maintaining quality and led to better preservation, which was not demonstrated in the tuna (probably due to the 20% initial glazing). Therefore, it could be assumed that the glazing could be reapplied during bulk sales depending on the type of fish, as it is a low-cost and time-saving practice. Moreover, it should be mentioned that these products normally never stay on sale for 60 days but have a shorter shelf life. This study has shown how re-glazing can have a positive effect on fish preservation, but looking to the future, one could also consider adding functional molecules with antimicrobial and antioxidant activity to the water used for re-glazing.

## Figures and Tables

**Figure 1 foods-14-01334-f001:**
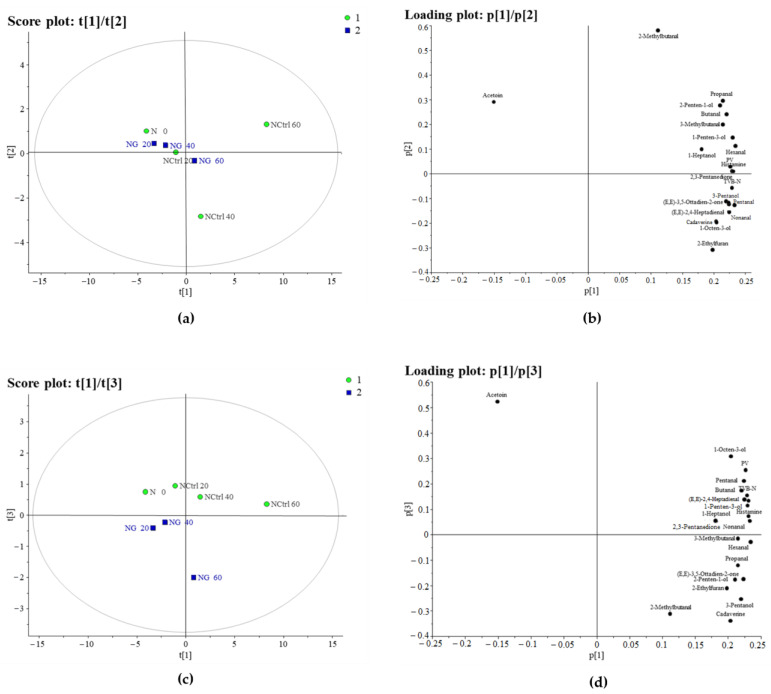
(**a**) Score (t1/t2) and (**b**) loading (p1/p2) plots of the first and second principal components (explaining 80% and 8%, respectively) of the PCA model, constructed using the hake samples. (**c**) Score (t1/t3) and (**d**) loading (p1/p3) plots of the first and third principal components (explaining 80% and 5%, respectively) of the PCA model, constructed using the hake samples. Green dots represent control samples, while blue squares represent glazed samples.

**Figure 2 foods-14-01334-f002:**
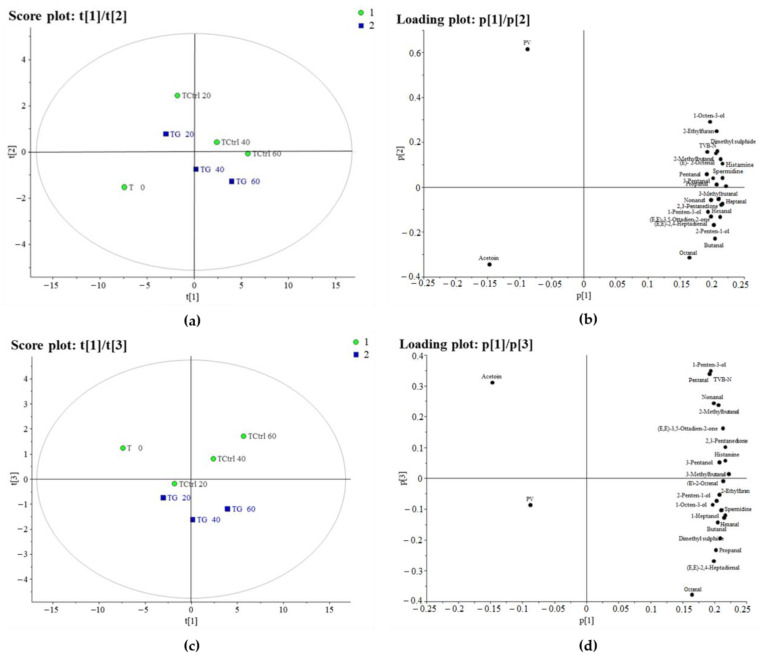
(**a**) Score (t1/t2) and (**b**) loading (p1/p2) plots of the first and second principal components (explaining 80% and 8%, respectively) of the PCA model, constructed using the tuna samples. (**c**) Score (t1/t3) and (**d**) loading (p1/p3) plots of the first and third principal components (explaining 80% and 7%, respectively) of the PCA model, constructed using the tuna samples. Green dots represent control samples, while blue squares represent glazed samples.

**Figure 3 foods-14-01334-f003:**
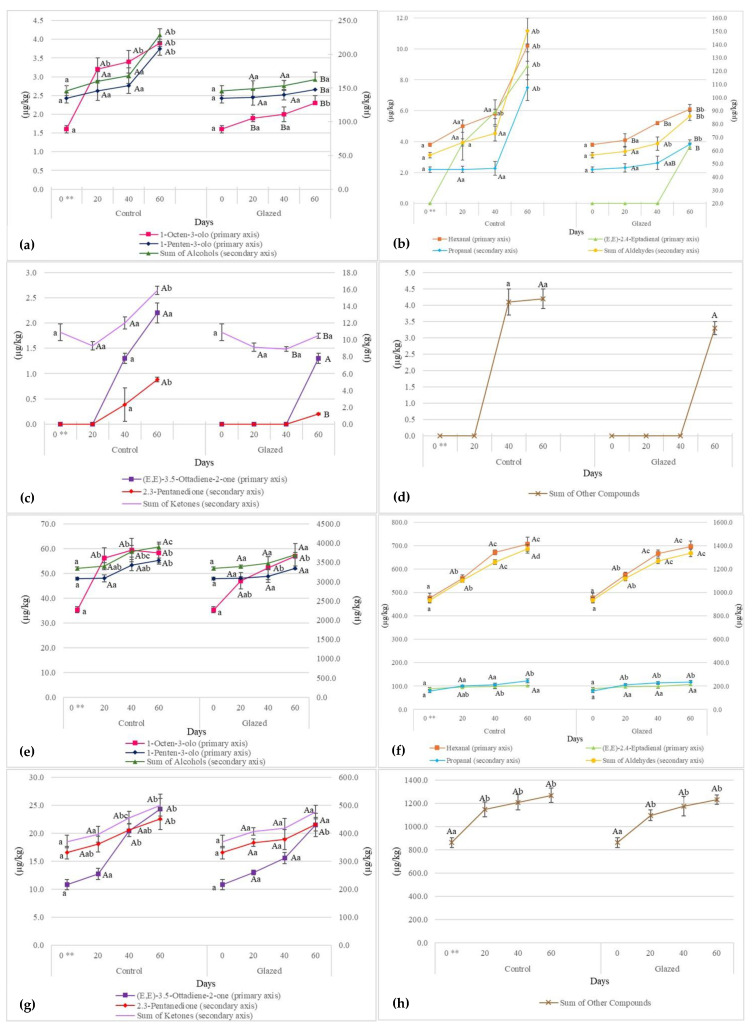
Evolution of volatile compounds—alcohols, aldehydes, ketones and other compounds (Dimethysulphide + 2-Ethylfuran) (µg/kg expressed as 4-methyl-2-pentanol)—of hake (**a**–**d**) and tuna (**e**–**h**). The results are the mean of two determinations ± the standard deviation. The comparison was performed using a *t*-test. Different lowercase letters indicate a statistically significant difference (*p* < 0.05) in the same sample during storage; different capital letters indicate a statistically significant difference (*p* < 0.05) between the two treatments at the same time. ** At T0, there is no capital letter because the samples are the same; therefore, no *t*-test was performed. T0 = 0 days; T1 = 20 days; T2 = 40 days; T3 = 60 days.

**Table 1 foods-14-01334-t001:** Evolution of the number of peroxides (meq of O_2_/kg) in hake and tuna *.

	Hake		Tuna	
Days	Control	Glazed	Control	Glazed
0 **	n.d.	n.d.	14.0 ± 0.4 a	14.0 ± 0.7 a
20	5.0 ± 0.1 a	n.d.	21.6 ± 1.1 Ab	21.9 ± 0.8 Ab
40	5.1 ± 0.2 a	n.d.	15 ± 0.5 Aa	13.8 ± 0.3 Aa
60	12.6 ± 0.3 Ab	2.6 ± 0.1 B	14.2 ± 0.4 Aa	11.9 ± 0.5 Ab

* The results are the mean of two determinations ± the standard deviation; n.d. indicates that the parameter could not be determined. The comparison was performed using a *t*-test. Different lowercase letters indicate a statistically significant difference (*p* < 0.05) in the same sample during storage; different capital letters indicate a statistically significant difference (*p* < 0.05) between the two treatments at the same time. ** At T0, there is no capital letter because the samples are the same; therefore, no *t*-test was performed. T0 = 0 days; T1 = 20 days; T2 = 40 days; T3 = 60 days.

**Table 2 foods-14-01334-t002:** Evolution of total volatile basic nitrogen (mg/100 g) in hake and tuna *.

	Hake		Tuna	
Days	Control	Glazed	Control	Glazed
0 **	16.8 ± 1.6 a	16.8 ± 1.4 a	20.3 ± 1.8 a	20.3 ± 1.6 a
20	20.3 ± 2 Aa	16.8 ± 1.3 Aa	22.4 ± 1.9 Aa	21 ± 1.9 Aa
40	21 ± 1.8 Aa	18.9 ± 1.5 Aa	23.8 ± 2.1 Aa	21.7 ± 2 Aa
60	24.5 ± 2.3 Aa	19.6 ± 1.7 Aa	25.2 ± 2 Aa	22.4 ± 2.1 Aa

* The results are the mean of two determinations ± the standard deviation. The comparison was performed using a *t*-test. Different lowercase letters indicate a statistically significant difference (*p* < 0.05) in the same sample during storage; different capital letters indicate a statistically significant difference (*p* < 0.05) between the two treatments at the same time. ** At T0, there is no capital letter because the samples are the same; therefore, no *t*-test was performed. T0 = 0 days; T1 = 20 days; T2 = 40 days; T3 = 60 days.

**Table 3 foods-14-01334-t003:** Evolution of biogenic amines (mg/kg) in hake and tuna *.

	Hake				Tuna			
Days	Control		Glazed		Control		Glazed	
	Cadaverine	Histamine	Cadaverine	Histamine	Spermidine	Histamine	Spermidine	Histamine
0 **	n.d.	n.d.	n.d.	n.d.	1.1 ± 0.1 a	4.1 ± 0.5 a	1.1 ± 0.9 a	4.1 ± 0.5 a
20	n.d.	n.d.	n.d.	n.d.	6.6 ± 0.5 Ab	9.5 ± 0.2 Ab	7.1 ± 0.4 Ab	8.8 ± 0.4 Ab
40	0.6 ± 0.05 Aa	0.4 ± 0.02 Aa	n.d.	n.d.	9 ± 1 Ac	11.5 ± 1 Ab	8.1 ± 1.5 Abc	9.9 ± 0.8 Ab
60	0.8 ± 0.1 Aa	1.0 ± 0.1 Ab	0.7 ± 0.04 A	0.2 ± 0.02 B	12.1 ± 0.2 Ad	15.7 ± 1.1 Ac	11.8 ± 1.0 Ac	12.3 ± 0.3 Ac

* The results are the mean of two determinations ± the standard deviation; n.d. indicates that the parameter could not be determined. The comparison was performed using a *t*-test. Different lowercase letters indicate a statistically significant difference (*p* < 0.05) in the same sample during storage; different capital letters indicate a statistically significant difference (*p* < 0.05) between the two treatments at the same time. ** At T0, there is no capital letter because the samples are the same; therefore, no *t*-test was performed. T0 = 0 days; T1 = 20 days; T2 = 40 days; T3 = 60 days.

## Data Availability

The original contributions presented in this study are included in the article. Further inquiries can be directed to the corresponding author.

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
