# Peer review of "A Simulation of the Real-Time Shelf Life of Frozen Fish Products in a Bulk System Sale"

_foods, 2025, doi:10.3390/foods14081334_

Round 1
Reviewer 1 Report
Comments and Suggestions for Authors
The paper investigated the real-time shelf-life of frozen fish fillets of two different fish species (tuna and cod) under simulated bulk marketing conditions, and explored the effect of ice-coat treatment on the quality of frozen fish. By simulating real-time sales conditions, the results of the study can provide a scientific basis for the storage and marketing of frozen fish products, and this research topic has important practical applications because shelf-life and quality maintenance of frozen fish products are key issues in the food industry. However, the following problems still exist:
- In the Introduction section, there is no space between numbers and ℃; in the Materials and Methods section, there is a space between numbers and letters, e.g., line 94; please check the whole text and correct it.
- In the Results and Discussion section, the indentation of the first line of all paragraphs should be consistent with the rest of the text.
- It is suggested to beautify the pictures of the data in the text to make them clearer and more intuitive.
- In the Introduction section, the potential advantages of ice-coating in extending the shelf-life of frozen fish and its uniqueness compared with other preservation methods can be further emphasized.
- When performing the ice-coating treatment, it is recommended to add the specific preparation method and composition of the ice-coating so that readers can have a clearer understanding of the experimental conditions.
- For the biogenic amine results, it is recommended to analyze in detail the mechanisms and sources of production of different biogenic amines, as well as their effects on fish flavor and safety.
Comments on the Quality of English Language
The English could be improved to more clearly express the research.
Reviewer 2 Report
Comments and Suggestions for Authors
Overall, the manuscript is well written and it is scientifically sound. The methodology is solid.
I have a few comments to improve in clarity, depth of discussion, and language the manuscript:
- It could be expressed with more emphasis the novelty of the study. The introduction could benefit with a discussion of how the research objectives are aligned with existing literature.
- Specify why certain storage conditions (e.g., -25°C, -18°C) were chosen and how they compare to real-world conditions.
- Clarify the rationale behind the selection of peroxide value, TVB-N, and biogenic amines as key indicators.
- Provide more details on the statistical validation methods used in PCA.
- In section 3.2, Peroxide Value, it is mentioned: “From T1 to T3, there was a decrease in PV in both cases, as was also to be expected according to Chaijan’s [24] previous report.” The authors could clarify whether the PV decrease is due to the degradation of primary oxidation products into secondary oxidation compounds (as measured in volatile analysis). Authors can support their affirmations by using what is reported in literature.
- In section 3.4, Biogenic Amines, it is mentioned: "The histamine content in the control was 0.4 mg/kg and 1 mg/kg, respectively, at 40 and 60 days, with a significant increase. In the glazed sample, only 0.2 mg/kg was measured after 60 days, and comparing it with the control, there was a significant difference, because the glaze reduced histamine production.” The claim that glazing reduced histamine formation lacks direct evidence explaining the mechanism. While histamine formation is often linked to microbial activity, freezing already inhibits bacterial growth.
- Please mention in Figure 1 the variance explained by each principal component
- The conclussion must directly address the practical implications of the findings, as well as possible future research directions.
There are occasional grammatical errors; a thorough proofreading or professional language review is recommended.
Round 2
Reviewer 1 Report
Comments and Suggestions for Authors
accept
Author Response
Thank you for accepting our revisions.
Reviewer 2 Report
Comments and Suggestions for Authors
In general, suggestions have been attended. However, I consider that still the methodology lacks of some details that could improve reproducibility. Providing more specifics on experimental conditions, data collection, and analysis techniques would strengthen scientific soundness.
Likewise, the conclusion could be more attached to the results than being instead a summary of the manuscript. More concise affirmations can be made.
Comments on the Quality of English LanguageSome sentences are unclear or overly complex, thus refining them would enhance readability.
Author Response
Comment 1: In general, suggestions have been attended. However, I consider that still the methodology lacks of some details that could improve reproducibility. Providing more specifics on experimental conditions, data collection, and analysis techniques would strengthen scientific soundness.
Response: Thank you for your comment. The experimental conditions and data collection are explained in sections 2.1, 2.2, 2.3, 2.4, 2.5. In particular, in section 2.3 we have tried to better describe how the glazing treatment is carried out, and in section 2.4 we have improved the description of the shelf-life trial. We also have revised the section 2.5, describing sampling and sample preparation for analysis. As regard methodology, section 2.9 on volatile compounds has been expanded to describe the analysis in more detail.
Comment 2: Likewise, the conclusion could be more attached to the results than being instead a summary of the manuscript. More concise affirmations can be made.
Response: Thank you for your suggestion. In the conclusions, on lines 545-560, we have included more details on the results obtained.
Comments on the Quality of English Language: Some sentences are unclear or overly complex, thus refining them would enhance readability.
Response: Thank you for pointing this out. We have already sent the manuscript to the MDPI’s Author Services for language editing, also after the major revisions. We attach the certificate again.
